# ON THE PRACTICALITY OF DETERMINISTIC EPISTEMIC UNCERTAINTY

## ABSTRACT

A set of novel approaches for estimating epistemic uncertainty in deep neural networks with a single forward pass has recently emerged as a valid alternative to Bayesian Neural Networks. On the premise of informative representations, these deterministic uncertainty methods (DUMs) achieve strong performance on detecting out-of-distribution (OOD) data while adding negligible computational costs at inference time. However, it remains unclear whether DUMs are well calibrated and can seamlessly scale to real-world applications - both prerequisites for their practical deployment. To this end, we first provide a taxonomy of DUMs, and evaluate their calibration under continuous distributional shifts. Then, we extend them to semantic segmentation. We find that, while DUMs scale to realistic vision tasks and perform well on OOD detection, the practicality of current methods is undermined by poor calibration under distributional shifts.

## 1 INTRODUCTION

Despite the dramatic enhancement of predictive performance of deep learning (DL), its adoption remains limited due to unpredictable failure on out-of-distribution (OOD) samples (1; 2) and adversarial attacks (3). Uncertainty estimation techniques aim at bridging this gap by providing accurate confidence levels on a model's output, allowing for a safe deployment of neural networks (NNs) in safety-critical tasks, *e.g.* autonomous driving or medical applications.

While Bayesian Neural Networks (BNNs) represent the predominant holistic solution for quantifying uncertainty (4; 5), exactly modelling their full posterior is often intractable, and scalable versions usually require expensive variational approximations (6; 7; 8; 9; 10). Moreover, it has recently been shown that true Bayes posterior can also lead to poor uncertainty (11). Thus, efficient approaches to uncertainty estimation largely remain an open problem, limiting the adoption within real-time applications under strict memory, time and safety requirements.

Recently, a promising line of work emerged estimating epistemic uncertainty in deterministic NNs with a single forward pass. By regularizing the hidden representations of a model, these methods represent an efficient and scalable solution to epistemic uncertainty estimation and to the related OOD detection problem. Compared to traditional uncertainty estimation techniques, Deterministic Uncertainty Methods (DUMs) quantify epistemic uncertainty by measuring distances (12; 13) or estimating the distribution of latent representations (14; 15; 16; 17; 18; 19; 20; 21). While OOD detection is a prerequisite for a safe deployment of DL in previously unseen scenarios, the calibration - *i.e.* how well uncertainty correlates with model performance - of such methods under continuous distributional shifts is equally important. Measuring the calibration of an epistemic uncertainty estimate on shifted data investigates whether it entails information about the predictive performance of the model. This an essential requirement for uncertainty and, unlike OOD detection, and evaluation that is not model-agnostic - i.e. one cannot perform well without taking the predictive model into account. Nonetheless, previous work falls short of investigating calibration, and solely focuses on OOD detection (12; 13; 16; 18; 19). Further, DUMs have thus far only been evaluated on toy datasets for binary classification, small-scale image classification tasks (13; 19) and toy prediction problems in natural language processing (15). Despite claiming to solve practical issues of traditional uncertainty estimation approaches, the practicality of DUMs remains to be assessed on more challenging tasks.

This work investigates the most crucial questions on the safety of DUMs in practical applications and addresses their shortcomings. In particular: (i) we provide the first analysis of the calibration

of DUMs under continuous distributional shifts; (ii) we evaluate the sensitivity of DUMs to their regularization strength; (iii) we scale DUMs to dense prediction tasks, *e.g.* semantic segmentation, and evaluate them under synthetic and realistic continuous distributional shift; (iv) we find that the practicality of many DUMs is undermined by their poor calibration under both synthetic and realistic distributional shifts.

## 2 RELATED WORK

**Sources of Uncertainty.** Uncertainty in a model's predictions can arise from two different sources (22; 23). While *aleatoric* uncertainty encompasses the noise inherent in the data and is consequently irreducible (22), *epistemic* uncertainty quantifies the uncertainty associated with choosing the model parameters based on limited information, and vanishes - in principle - in the limit of infinite data. This work distinguishes two properties of epistemic uncertainty - its performance on detecting OOD samples and its calibration (i.e. its correlation with model performance under distributional shifts). While the latter has been explored in the case of probabilistic approaches to uncertainty estimation (24) we are the first to investigate the behaviour of DUMs in this scenario. Notably, (25) evaluates prominent scalable epistemic uncertainty estimates on semantic segmentation. However, they investigate calibration only on in-distribution data. Further, although (15) evaluates the calibration of their approach, they do so exclusively on in-distribution data.

**BNNs** (26; 5) represent a principled way of measuring uncertainty. However, their intractable posterior distribution requires approximate inference methods, such as Markov Chain Monte-Carlo (26) or Variational Inference (VI) (4). While these methods traditionally struggle with large datasets and architectures, a variety of scalable approaches - often based on VI - have recently emerged.

**Deep Ensembles**, which typically consist of identical models trained from different initializations, have been introduced to the deep learning community by Lakshminarayanan *et al.* (27) and extended by (28; 29). While deep ensembles are widely regarded as a strong baseline for estimating epistemic uncertainty, they come with high computational as well as memory costs.

**Efficient Approaches.** Recently, approaches based on stochastic regularization have been developed (6; 7; 8; 30; 31). By keeping stochasticity at inference time, they estimate uncertainty using multiple forward passes. Another line of work estimates the posterior using the Laplace-approximation (32; 33; 34) Moreover, efficient ensemble methods were proposed producing predictions using a single model (35; 29; 28; 36; 37). Despite promising results on large-scale tasks while being parameter-efficient, these methods still require sampling through the model, which can render them impractical given limited compute. To estimate uncertainty in real-time and resource-demanding tasks, recent work has focused on providing uncertainty estimates with a *single forward pass*. One line of work proposes a principled approach for variance propagation in NNs (9; 38; 10). Notably, another line of work proposes efficient approaches to estimate aleatoric uncertainty (39; 40; 41).

Recently, DUMs showed promising results on OOD detection. By leveraging distances and densities in the feature space of a NN, these methods provide confidence estimates while adding negligible computational cost. Since they are united in their deterministic treatment of the weights, we term them Deterministic Uncertainty Methods (DUMs). The next section provides a taxonomy of DUMs.

## 3 TAXONOMY FOR DETERMINISTIC UNCERTAINTY QUANTIFICATION

This section categorizes existing DUMs. To quantify epistemic uncertainty deterministically, the distribution of the hidden representations of a NN needs to be sensitive to the input distribution. However, discriminative models suffer from the fundamental problem of feature collapse (13; 19). Thus, we firstly categorize DUMs according to the regularization method used to counteract feature collapse (Sec. 3.1). Moreover, we cluster DUMs according to the method used for uncertainty estimation (Sec. 3.2). Tab. 1 shows an overview of the resulting taxonomy.

**Feature Collapse.** Discriminative models can learn to discard large part of their input information, as exploiting spurious correlations may lead to better performance on the training data distribution (46; 47). Such invariant representations learned may be blind to distributional shifts, resulting in a collapse of OOD embeddings to in-distribution features This problem is known as *feature collapse* (13), and it makes OOD detection based on high-level representations impossible.

| **DUMs** | | **Uncertainty Estimation Method** | | | |
| --- | --- | --- | --- | --- | --- |
| | | Discriminative | | Generative | |
| | | Distance from class centroid | Gaussian Processes | Gaussian Mixture Models | Normalizing Flows |
| **Regu-larization** | Distance awareness | DCS (12), DUQ (13) | SNGP (15), DUE (18) | DDU (19) | - |
| | Informative representations | - | - | DCU (16; 42), MIR (20) | Invertible networks (43; 44; 45) PostNet(17) |

Table 1: Taxonomy of DUMs. Methods are grouped according to their regularization of the hidden representations (rows), and their uncertainty estimation method (columns).

## 3.1 REGULARIZATION OF HIDDEN REPRESENTATIONS

We group DUMs according to how feature collapse is tackled. We identify two main paradigms - distance awareness and informative representations - which we discuss in Sec. 3.1.1 and Sec. 3.1.2.

### 3.1.1 DISTANCE AWARENESS

Essentially, distance-aware hidden representations avoid feature collapse by relating distances between latent representations to distances in the input space. Therefore, one constrains the bi-Lipschitz constant, as it enforces a lower and an upper bound to expansion and contraction performed by a model. A lower bound enforces that different inputs are mapped to distinct representations and, thus, provides a solution to feature collapse. The upper bound enforces smoothness, *i.e.* small changes in the input do not result in large changes in the latent space. While there exist other approaches, *e.g.* (48), recent proposals have primarily adopted two methods to impose the bi-Lipschitz constraint.

The two-sided **Gradient Penalty** relates changes in the input to changes in feature space by directly constraining the gradient of the input (13). **Spectral Normalization (SN)** (49) is a less computationally-demanding alternative. SN is applicable to residual layers and normalizes the weights $W$ of each layer using their spectral norm $sn(W)$ to constrain the bi-Lipschitz constant. Various DUMs - SNGP (15), DUE (18) and DDU (19) - rely on SN to enforce distance-awareness of hidden representations. More details on gradient penalty and SN can be found in the supplement.

Note that the Lipschitz constraint is defined with respect to a fix distance measure, which can be difficult to choose for many high-dimensional data distributions. A popular choice, e.g. (13; 15; 18), is the $L_2$ distance. Moreover, principled approaches to provide exact singular values in convolutional layers (50) result in prohibitive computational complexity; the spectral normalization approximations typically adopted by the methods previously described have been found to be sub-optimal (51), and its interaction with losses, architecture and optimization is yet to be fully understood (52).

### 3.1.2 INFORMATIVE REPRESENTATIONS

While distance-awareness achieves remarkable performance on OOD detection, it does not explicitly preserve sample-specific information. Thus, depending on the underlying distance metric it may discard useful information about the input or act overly sensitive. An alternative line of work avoids feature collapse by learning informative representations (14; 16; 20; 44; 43; 45), thus forcing discriminative models to preserve information in its hidden representations beyond what is required to solve a task independent of the choice of an underlying distance metric. Notably, while representations that are aware of distances in the input space are also informative, both categories remain fundamentally different in their approach to feature collapse. While distance-awareness is based on the choice of a specific distance metric tying together input and latent space, informative representations incentivize a NN to store more information about the input using an auxiliary task (20; 16) or forbid information loss by construction (43; 44; 45). We identify three distinct families of approaches.

**Contrastive learning** (53) has emerged as an approach for learning representations that are both informative and discriminative. This is utilized by Wu *et al.* (16) and Winkens *et al.* (42), who apply

SimCLR (54) to regularize hidden representations for a discriminative task by using a contrastive loss for pretraining and fine-tuning to force representations to discriminate between individual instances.

**Reconstruction regularization** (20) instead forces the intermediate activations to fully represent the input. This is achieved by adding a decoder branch fed with the activations of a given layer to reconstruct the input. We term it MIR (Maximally Informative Representations).

**Entropy regularization.** PostNet (17) learns the class-conditional distribution of hidden representations end-to-end using a Normalizing Flow (NF) parameterizing a Dirichlet distribution. This allows them to enforce informative representations by implicitly encouraging large entropy of the NF during training. We refer to the supplement for details.

**Invertible Neural Networkss (INNs)** (55; 43; 44; 45), built via a cascade of homeomorphic layers, cannot discard information except at the final classification stage. Consequently, the mutual information between input and hidden representation is maximized by construction. Interestingly, Behrmann *et al.* (56) showed that a ResNet is invertible if its Lipschitz constant is lower than 1, meaning that invertible ResNets both possess highly-informative representations and satisfy distance-awareness. However, note that this is not a necessary condition for invertibility, and thus information preservation.

## 3.2 Uncertainty Estimation

We distinguish two directions regarding uncertainty estimation in DUMs - generative and discriminative approaches. While generative approaches use the likelihood produced by an explicit generative model of the distribution of hidden representations as a proxy for uncertainty, discriminative methods directly use the predictions based on regularized representations to quantify uncertainty.

**Generative approaches** estimate the distribution of hidden representations post-training and use the likelihood as uncertainty metric. Wu *et al.* (16) propose a method to estimate the distribution in the feature space, where the variance of the distribution is used as a confidence measure. MIR (20), DDU (19) and DCU (42) fit a class-conditional GMM to their regularized hidden representations and use the log-likelihood as an epistemic uncertainty proxy. DEUP (57) uses the log-likelihood of a normalizing flow in combination with an aleatoric uncertainty estimate to predict the generalization error. A special instance of the generative approaches are INNs as they directly estimate the training data distribution. The likelihood of the input data is used as a proxy for uncertainty. While this idea is appealing, it can lead to training difficulties, imposes strong constraints on the underlying model and still remains susceptible to OOD data (58). PostNet (17) is a hybrid approach which estimates the distribution of hidden representations of each class using a separate NF end-to-end. Its log-likelihoods parameterize a Dirichlet distribution. We categorize PostNet as a generative approach since their epistemic uncertainty is the log-likelihood of the NF associated with the predicted class.

**Discriminative approaches** use the predictions to directly assess confidence. Mandelbaum *et al.* (12) propose to use a Distance-based Confidence Score (DCS) learning a centroid for each class end-to-end. Similarly, DUQ (13) builds on Radial Basis Function (RBF) networks (59) and proposes a novel centroid updating scheme. Both estimate uncertainty as the distance between the model output and the closest centroid. DUMs adopting SN (15; 18) (preserving $L_2$ distances) typically replace the softmax layer with Gaussian processes (GPs) with RBF kernel, extending distance awareness to the output layer. In particular, SNGP (15) relies on a Laplace approximation of the GP based on the random Fourier feature (RFF) expansion of the GP posterior (60). DUE (18) uses the inducing point approximation (61; 62), incorporating a large number of inducing points without overfitting (63). The uncertainty is derived as the Dempster-Shafer metric (15), resp. the softmax entropy (18).

# 4 Evaluation of Deterministic Epistemic Uncertainty

We measure the calibration of DUMs under continuous distributional shifts in two parts. Firstly, we evaluate DUMs on image classification using synthetic corruptions (Sec. 4.1.1). Then, we extend DUMs to a large-scale dense prediction task - semantic segmentation (Sec. 4.2) - where we evaluate their calibration on synthetic corruptions (Sec. 4.2.1) based on Cityscapes and on more realistic distributional shifts (Sec. 4.2.2) based on data collected in the simulation environment CARLA (64).

**Baselines.** We compare DUMs with two baselines for epistemic uncertainty - Monte-Carlo (MC) dropout (7) and deep ensembles (27). Moreover, we report the softmax entropy of a vanilla NN as a

simple baseline. We refer to the supplement for details on uncertainty estimation in our baselines. Note, that the softmax entropy is expected to yield suboptimal calibration under distributional shifts since it quantifies aleatoric uncertainty while adding no computational overhead.

**Methods.** We evaluate DUQ (13), SNGP (15) and DDU (19) as representatives of distance-awareness, since these cover both techniques - SN and gradient penalty - and apply different techniques for uncertainty estimation. We exclude DUE (18) since it provides limited additional insights given SNGP. Moreover, we exclude DCS (12) since it only leads to a marginal improvement in their own experiments and their contrastive loss only operates on class centroids and, thus, is not expected to lead to distance awareness within clusters. Furthermore, we evaluate MIR (20), DCU (42) and PostNet (17) as representatives of informative representations. However, we do not scale DCU and PostNet to semantic segmentation. DCU with its constrastive pretraining based on SimCLR (54) is computationally too demanding due to large batch sizes. PostNet does not scale to semantic segmentation due instabilities arising from the end-to-end training of the NF for learning the distribution of hidden representations. They require a small hidden dimension (<10) which already leads to poor testset performance on CIFAR100. Further, we do not evaluate methods based on invertible neural networks (44; 45) since they 1) enforce strict constraints on the underlying architecture (*e.g.* fixed dimensionality of hidden representations) and often lead to training instabilities.

**Calibration Metrics.** Typical calibration metrics are Expected Calibration Error (ECE) (65) and Brier score (66). However, since most DUMs, except SNGP, do not provide uncertainty in form of a probabilistic forecast, we cannot rely on measuring the calibration of probabilities. Thus, we will exploit another desired property of uncertainty to quantify calibration, namely the ability to distinguish correct from incorrect predictions. In fact, this property is a relaxation of calibrated probabilities, as it is independent of the absolute value of uncertainty estimates and only relies on the ordering of predictions relative to each other. We assess the calibration of uncertainty estimates under distributional shifts using two metrics. Firstly, we report the *Area Under the Receiver Operating Characteristic (AUROC)* obtained when separating correct and incorrect predictions based on uncertainty. Moreover, we introduce a new metric, *Relative Area Under the Lift Curve (rAULC)*, based on the Area Under the Lift Curve (AULC) (67). The AULC is obtained by ordering the predictions according to increasing uncertainty and plotting the performance of all samples with an uncertainty value smaller than a certain quantile of the uncertainty against the quantile itself.

Formally, given a set of uncertainty quantiles $q_i \in [0, 1]$, $i \in [1, ..., S]$, with some quantile step width $0 < s < 1$ and the function $F(q_i)$ which returns the accuracy of all samples with uncertainty $u < q_i$, the AULC is defined as $AULC = -1 + \sum_{i \in [1, ..., S]} s \frac{F(q_i)}{F_R(q_i)}$. Here, $F_R(\cdot)$ refers to a baseline uncertainty estimate that corresponds to random guessing, and we subtract 1 to shift the performance of the random baseline to zero. Note, if an uncertainty estimate is anti-correlated with a models' performance, this score can also be negative. To alleviate a bias towards better performing models, we further compute the rAULC by dividing the AULC by the AULC of a hypothetical (optimal) uncertainty estimation that perfectly orders samples according to model performance. In classification we measure AUROC and rAULC on the image-level, in semantic segmentation on the pixel-level. In all experiments we set the quantile step width to $s = \frac{1}{N}$, where N is the number of predictions.

We compute AUROC and rAULC on continuous distributional shifts 1) across all severities of distributional shifts (including the clean testset) and 2) for each severity separately. We use the former method to establish a quantitative comparison among the methods and the latter to depict the calibration evolution qualitatively as a function of the distributional shift's severity.

## 4.1 IMAGE CLASSIFICATION

**Datasets.** We train DUMs on CIFAR-10 and CIFAR-100 (68) and evaluate on the corrupted versions of their test set CIFAR10/100-C (69) (Sec. 4.1.1). These include 15 synthetic corruptions, each with 5 levels of severity. Moreover, we explore how sensitive the calibration of DUMs is to the choice of regularization strength on MNIST (70) and FashionMNIST (71)).

**Models and optimization.** Each method shares the same backbone architecture and uses a method-specific prediction head. When training on CIFAR-10/100, the backbone architecture is a ResNet-50 (72) For the experiments regarding hyperparameter sensitivity on MNIST and Fashion-MNIST, we employ a multilayer perceptron (MLP) as feature extractor with 3 hidden layers of 100 dimensions each and ReLU activation functions. Each DUM has a hyperparameter for the regularization of its

hidden representations. We choose the hyperparameter such that it minimizes the validation loss. All results are averaged over 5 independent runs. The standard deviation and optimization details can be found in the supplement where not present in the main paper.

### 4.1.1 CONTINUOUS DISTRIBUTIONAL SHIFTS

| Method | CIFAR10-C | | | CIFAR100-C | | |
|---|---|---|---|---|---|---|
| | ACC | AUROC | rAULC | ACC | AUROC | rAULC |
| Softmax | 0.882 | 0.782 | 0.708 | 0.610 | 0.762 | 0.596 |
| MC Dropout (7) | 0.885 | **0.866** | 0.829 | 0.615 | 0.818 | **0.726** |
| Ensemble (27) | 0.910 | 0.85 0 | **0.833** | 0.628 | **0.824** | 0.713 |
| SNGP (15) | 0.903 | 0.833 | 0.766 | 0.611 | 0.788 | 0.623 |
| DDU (19) | 0.884 | 0.673 | 0.441 | 0.609 | 0.635 | 0.339 |
| MIR (20) | 0.889 | 0.79 | 0.697 | 0.617 | 0.726 | 0.514 |
| DUQ (13) | 0.860 | 0.773 | 0.614 | - | - | - |
| DCU (42) | **0.945** | 0.794 | 0.706 | **0.642** | 0.750 | 0.558 |
| PostNet (17) | 0.882 | 0.784 | 0.676 | 0.520 | 0.743 | 0.603 |

Table 2: We compare Softmax, MC Dropout (7), Deep Ensembles, SNGP, DDU, MIR, DUQ, DCU and PostNet on CIFAR10/100-C. We evaluate the accuracy (ACC) on the uncorrupted testset, AUROC and rAULC. Ensembles and MC dropout demonstrate better uncertainty calibration than most DUMs. Only SNGP consistently outperforms the softmax entropy. DCU's superior performance is expected since it uses expensive contrastive pretraining. DUQ did not converge on CIFAR100-C due to training instabilities arising from dynamically updated cluster centroids.

Tab. 2 reports testset accuracy and calibration for the baselines and DUMs. AUROC and rAULC are computed for each corruption across all severity levels, then averaged over all corruptions. Further, Fig. 1 depicts our metrics depending on the severity of corruptions. We generally observe that ensembles and MC dropout demonstrate best performance in terms of calibration. Further, SNGP is the only DUM that consistently outperforms the softmax entropy. Overall we observe that DUMs using the distribution of hidden representations for estimating epistemic uncertainty yield worse calibration. Among these we find that regularizing hidden representations by enforcing distance awareness (DDU) yields the worst calibration. The superior performance of DCU (42) in terms of testset accuracy originates from their extensive contrastive pretraining which includes extensive data augmentation and a prolonged training schedule. We note that DUQ (13) did not converge on CIFAR100 due to training instabilities. These arise from maintaining the class centroids, which become very noisy for 100 classes with only 600 samples per class.

Moreover, we report OOD detection performance of models used in Tab. 2 and Fig. 1 in the supplement. Interestingly, despite competitive performance of OOD detection DUMs fall short in terms of uncertainty calibration compared to MC dropout and deep ensembles.

### 4.1.2 SENSITIVITY TO HYPERPARAMETERS

We are interested in the qualitative impact of the regularization strength on the uncertainty calibration. Therefore, we train DUQ (13), SNGP (15), MIR (20) and DDU (19) using various regularization strengths on MNIST and evaluate on continuously shifted data by rotating from 0 to 180 degrees in steps of 20 degrees. Fig. 2 depicts the test accuracy against the rAULC for various regularization strengths. Only for MIR we observe a clear, positive correlation between regularization strength and calibration. The supplement depicts similar results on FashionMNIST (71).

### 4.2 SEMANTIC SEGMENTATION

This section evaluates whether DUMs seamlessly scale to realistic vision tasks and compares their behaviour under synthetic and realistic continuous distributional shifts with the softmax entropy, MC dropout and ensembles. Therefore, we apply MIR (20), SNGP (15) and DDU (19) to semantic segmentation. Note that DUQ (13) did not converge on this task.

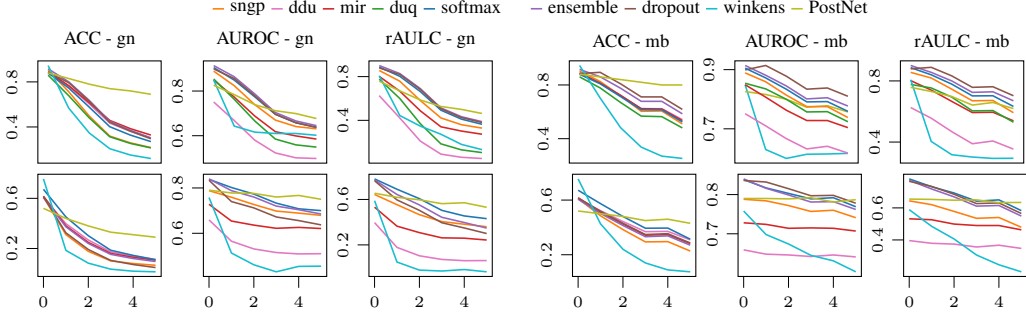

Figure 1: Softmax entropy, ensembles (27), MC dropout (7), DUQ (13), SNGP (15), MIR (20), DDU (19) and DCU (42) on CIFAR10-C (upper row) and CIFAR100-C (lower row) (69). We show the accuracy, AUROC and rAULC on the corruptions gaussian_noise (gn) and motion_blur (mb) against the corruption severity. While all methods, except DCU, demonstrate a similar accuracy, DUMs - in particular methods based on generative modeling of hidden representations - yield worse calibration. DUQ did not converge on CIFAR100. Other corruptions are included in the supplement.

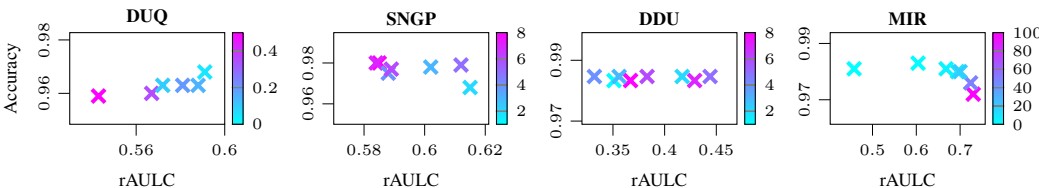

Figure 2: We analyse the sensitivity of DUQ (13), SNGP (15), MIR (20) and DDU (19) to their regularization strength. Therefore, we train models on MNIST and evaluate on continuously shifted data by rotating from 0 to 180 degrees in steps of 20 degrees. We plot the accuracy on the unperturbed testset against the rAULC computed using data from all levels of perturbation. Only for MIR we observe a clear, positive correlation between regularization strength and calibration. For DDU and SNGP, smaller regularization parameter denotes stronger regularization.

We consider semantic segmentation as a multidimensional classification problem, where each pixel of the output mask represents an independent classification problem. Given an image $\mathbf{x}$ with $n$ pixels $\mathbf{y} = \{y_1, \cdots, y_n\}$, the predictive distribution factorizes according to $p(\mathbf{y} \mid \mathbf{x}) = p(y_1 \mid \mathbf{x})p(y_2 \mid \mathbf{x}) \cdots p(y_n \mid \mathbf{x})$. We evaluate the calibration of the pixel-level uncertainty in our experiments.

**Datasets.** We evaluate on synthetic distributional shifts using a corrupted version of Cityscapes (73) (Cityscapes-C (74)) which contains the same corruptions as CIFAR10/100-C. To further benchmark DUMs in a realistically and continuously changing environment, we collect a synthetic dataset for semantic segmentation. We use the CARLA Simulator (64) for rendering the images and segmentation masks. The classes definition is aligned with the CityScape dataset (73). Training data is collected from four towns in CARLA. We produce 32 sequences from each town. Vehicles and pedestrians are randomly generated for each sequence. Every sequence has 500 frames with a sampling rate of 10 FPS. We uniformly sample a validation set. We introduce continuous distributional shifts by varying the time-of-the-day and weather conditions (visual examples and details on data collection are in the supplment). The time-of-the-day is parameterized by the sun's altitude angle, where $90°$ means mid-day (training data) and the $0°$ means dust/dawn. We produce samples with altitude angles from $90°$ to $15°$ by steps of $5°$, and $15°$ to $-5°$, where the environment changes sharply, in $1°$ steps. In order to continuously change the weather conditions, we increase the magnitude of the rain in four steps (see supplement for visual examples). We refer to this dataset as CARLA-C.

**Backbone.** We adopt Dilated ResNet (DRN) (75; 76) as semantic segmentation backbone since it is based on residual connections allowing the use SN. Using dilated convolutions it improves spatial accuracy, achieving satisfactory results on CityScapes (73). We adopt the variant DRN-A-50. All results are averaged across 5 independent repetitions.

**SNGP.** DRN uses $1 \times 1$ convolutions at the last layer to map the latest feature map to the predicted segmentation mask. This works under the assumption that all pixels in the output mask are i.i.d.

random variables. Following this intuition, we extend SNGP to semantic segmentation by fitting a $GP : \mathbb{R}^Z \rightarrow \mathbb{R}^C$ at pixel level that maps from the deep feature dimension $z$ to the number of classes $c$. By keeping the GP kernel parameters shared across all pixels, we simulate a $1 \times 1$ convolutional GPs, *i.e.* $\sigma : (H_l \times W_l \times Z) \rightarrow (H_l \times W_l \times C)$, where $\sigma$ convolves the GP, $H_l$ and $W_l$ are, respectively, feature map height and width at layer $l$, $Z$ is the number of latent features and $C$ is the number of output classes. For details about the GP we refer to (15) or the supplement.

**MIR and DDU** require fitting the distribution of hidden representations. We fit a Gaussian mixture model (GMM) with 20 components (*i.e.* number of classes) to each spatial location of the hidden representations using features extracted from the training data independently. This assumes that the distribution is translation invariant and factorizes along the spatial dimensions of the latent space. Pixel-level uncertainties are then computed using bi-cubic interpolation following a similar procedure as (77) in this framework. We refer to the supplement for more details

### 4.2.1 CITYSCAPES CORRUPTED

We evaluate the softmax entropy, ensembles (27), MC dropout (7), SNGP (15), MIR (20), DDU (19) on Cityscapes-C (74). Tab. 3 and Fig. 3 depict mean Intersection over Union (mIoU) and calibration performance in terms of AUROC and rAULC. Ensembles and MC dropout yield the best calibration, while among DUMs only SNGP consistently ourperforms the softmax entropy.

| Method | Cityscapes-C | | | CARLA-C | | |
|---|---|---|---|---|---|---|
| | mIoU | AUROC | rAULC | mIoU | AUROC | rAULC |
| Softmax | 0.503 | 0.815 | 0.737 | 0.422 | 0.854 | 0.818 |
| MC Dropout (7) | 0.506 | **0.846** | **0.785** | 0.410 | 0.843 | 0.730 |
| Ensemble (27) | 0.525 | 0.835 | 0.751 | **0.428** | **0.863** | 0.812 |
| SNGP (15) | **0.519** | 0.833 | 0.759 | 0.424 | 0.853 | **0.813** |
| DDU (19) | 0.505 | 0.731 | 0.542 | 0.408 | 0.467 | -0.038 |
| MIR (20) | 0.504 | 0.729 | 0.564 | 0.412 | 0.744 | 0.619 |

Table 3: We compare semantic segmentation using Softmax, MC Dropout (7), Deep Ensembles, SNGP, DDU and MIR on Cityscapes-C and CARLA-C. We evaluate the mIoU on the uncorrupted testset and AUROC/rAULC across all levels of corruption. Again, ensembles and MC dropout yield better calibrated uncertainty than most DUMs. Most notably, only SNGP consistently outperforms the softmax entropy. DUMs using an explicit generative model of hidden representations to estimate uncertainty perform particular bad on realistic distributional shifts (CARLA-C).

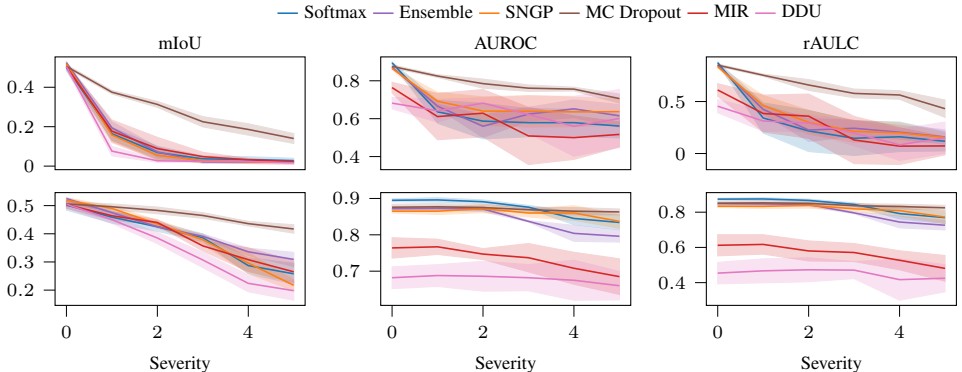

Figure 3: Softmax entropy, ensembles (27), MC dropout (7), SNGP (15), MIR (20), DDU (19) on Cityscapes-C (74). We show the mIoU, AUROC and rAULC on the corruptions gaussian_noise (upper) and motion_blur (lower) depending on the corruption severity. While all methods demonstrate a similar mIoU, DUMs - in particular methods based on generative modeling of hidden representations - yield worse calibration across corruption severities. Other corruptions are in the supplement.

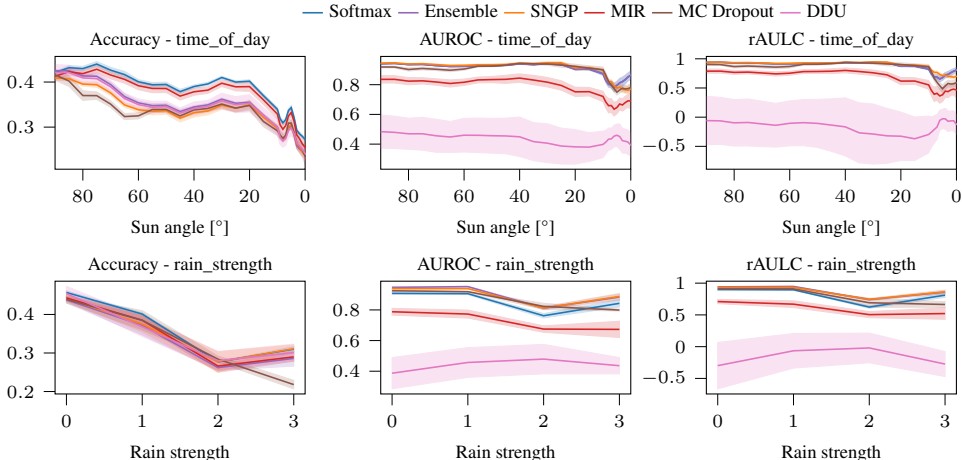

Figure 4: Softmax entropy, ensembles (27), MC dropout (7), SNGP (15), MIR (20), DDU (19) on CARLA-C (74). We show the mIoU, AUROC and rAULC for each method on the corruptions "time of day" (upper) and "rain" (lower) depending on the corruption severity. While all methods demonstrate a similar mIoU pattern, DUMs - in particular methods based on generative modeling of hidden representations - yield worse calibration across corruption severities.

### 4.2.2 REALISTIC CONTINUOUS DISTRIBUTIONAL SHIFTS

Similarly, we evaluate the softmax entropy, ensembles (27), MC dropout (7), SNGP (15), MIR (20), DDU (19) on CARLA-C. Tab. 3 and Fig. 4 depict mIoU and calibration performance in terms of AUROC and rAULC. Ensembles yield the best calibration. Among DUMs only SNGP consistently ourperforms the softmax entropy which is in line with the results on image classification (Sec. 4.1.1).

## 5 CONCLUSION & DISCUSSION

This work investigated the calibration under continuous distributional shifts of DUMs which recently showed good OOD detection performance and are interesting for practical applications in need of efficient uncertainty quantification. Overall, we observe that such uncertainty estimates are considerably worse calibrated than scalable Bayesian methods. We observe this on image classification (Sec. 4.1.1) as well as semantic segmentation (Sec. 4.2) and on synthetic as well more realistic distributional shifts. SNGP (15) denotes the only DUM that consistently yields better calibrated uncertainties under continuous distributional shifts than the softmax entropy. Simultaneously, SNGP is the only DUM which derives there uncertainty from its predictive distribution.

In particular, our experiments reveal that methods relying on the distribution of hidden representations to quantify uncertainty (42; 20; 19) are poorly calibrated. Arguably, it is understandable that these methods are worse calibrated than SNGP since they do not take into account the predictive distribution. The underlying assumption is that locations in feature space entail information about the correctness of predictions. While this is arguably true, features also contain additional information that render them suboptimal for judging the correctness of predictions due to ambiguities.We refer to the supplement for a further theoretical consideration of the limitations of DUMs. Overall, this underlines the necessity to refrain from DUMs that purely rely of distance or log-likelihood in the feature space when well calibrated uncertainties are required.

Moreover, another desirable property for such methods would be that the strength of the feature space regularization correlates with the quality of the uncertainty. Due to the original purpose of most DUMs, this would be at least expected for OOD detection. However, this is not verified for Lipschitz regularization by our investigation (Sec. 4.1.1). We hypothesize that this originates from the choice of metric for regularization - *i.e.* $L_2$ distance - which is not meaningful in the image space. We hope that our findings will foster future research on making these promising family of methods better calibrated and more broadly applicable.

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
