# OpenReview forum: "On the Practicality of Deterministic Epistemic Uncertainty"
_ICLR.cc/2022/Conference — ICLR 2022 Submitted_

### Official Review · Reviewer_JspH · 2021-10-31

**Correctness:** 3
**Technical Novelty And Significance:** 1
**Empirical Novelty And Significance:** 2
**Recommendation:** 5
**Confidence:** 4

**Main Review:**

The reasons invoked to include some of the cited DUM approaches are not very convincing, since the point of a benchmark is to evaluate all the relevant approaches according to the same yardstick. Since there is no methodological novelty in this paper, I would suggest to open up the comparison to a slightly larger set of methods than what was done here.

Nonetheless, the provided comparisons can be useful for future research aiming at proposing novel approaches and possibly avoiding all the possible comparisons, focusing instead on the methods which seem to perform well in this paper. These comparisons also point to DUM approaches being generally insufficiently well calibrated (especially those relying on the distribution of hidden representations).

There is another deterministic estimator of epistemic uncertainty that could have been considered, of a very different nature (but only applicable with one can have an estimator of aleatoric uncertainty, e.g. for pairs of examples with the same x but different y): DEUP (Lahlou et al, 2021). They train a 2nd network to predict the per-example out-of-sample error of the main predictor.

**Summary Of The Paper:**


This paper reviews and compares a class of epistemic uncertainty estimation methods that avoid sampling (hence called deterministic) as well as multiple models (like ensembles) because of their memory footprint.

**Summary Of The Review:**


This paper compares experimentally a number of epistemic uncertainty estimators within a fairly limited but interesting class of methods. This could be useful to practitioners of the field. A broader set of comparisons could improve the paper.

---

> ### Author Response · Authors · 2021-11-15
> **Response to JspH**
>
> **Scope of this work.**
>
> We did not intentionally choose a narrow circle of methods, but rather started from the observation that there is this novel set of approaches that treats the weights of a neural network deterministically while claiming to estimate epistemic uncertainty, which lacks theoretical guarantees for good uncertainty estimates.  Currently the field of DUMs is still very young and scattered (in particular when considering works published at top tier conferences). Thus, there is naturally only a limited number of works suitable for this comparison. Note that we further include Dirichlet-based methods into our analysis (see response to ZZs6).
>
> **Direct epistemic uncertainty**
>
> We thank the reviewer for pointing out this recent unpublished work. We added it to our taxonomy section.

---

### Official Review · Reviewer_ZZs6 · 2021-11-01

**Correctness:** 3
**Technical Novelty And Significance:** 1
**Empirical Novelty And Significance:** 3
**Recommendation:** 5
**Confidence:** 4

**Main Review:**

Strengths:

(1)	The paper is well-organized. It first introduces a summary of DUMs and then provides experiments on both image classification tasks and real-world applications, i.e., semantic segmentation.

(2)	The experiments are systematic, including all the necessary evaluation metrics and different levels of distribution shifts.

Weaknesses:

For the method:

(1)	Some important deterministic uncertainty methods such as the Dirichlet-based methods [1,2,3] are missing in the paper.  Moreover, the authors cite the Posterior network [2] work as one of the DUMs but did not evaluate it.  In addition, [4], that performed the evaluation of the Dirichlet-based methods , also considered the uncertainty robustness under distribution shifts. The authors should provide a discussion and comparison of their work with [4].

(2) The authors should discuss in the body of the papers the about underlying reasons for the issues associated with the DUM methods and even propose possible solutions to address or avoid these problems.

(3)	The authors should discuss each deterministic uncertainty method in terms of the strengths/weaknesses of each method and the possible assumptions. Some necessary equations should also be provided. Then, we can have a clear mind about which method to choose for experiments

(4)	The authors mainly evaluate the uncertainty robustness under distributional shifts. However, the author should also provide some discussion about the OOD detection results. Could DUMs perform as well as Ensemble and MC-dropout when there are no distributional shifts?

For the experiments

(1)	The experiment results lack the necessary analysis. For example, why do most DUMs have much worse calibration performance than softmax?

(2) 5 DUMs are considered for image classification while only 3 DUMs are considered for semantic segmentation. The performance of DUMs may not be representative of semantic segmentation.

(3)	The experiments did not consider the regression tasks.

(4)	Figures 1,2,3,4 are very confusing. It is hard to distinguish different colors from figures. The authors may use different curves when plotting the results to distinguish DUMs with Ensemble and MC-dropout.

[1] Malinin, Andrey, and Mark Gales. "Predictive uncertainty estimation via prior networks." arXiv preprint arXiv:1802.10501 (2018).

[2] Charpentier, Bertrand, Daniel Zügner, and Stephan Günnemann. "Posterior network: Uncertainty estimation without ood samples via density-based pseudo-counts." arXiv preprint arXiv:2006.09239 (2020).

[3] Malinin, Andrey, Bruno Mlodozeniec, and Mark Gales. "Ensemble distribution distillation." arXiv preprint arXiv:1905.00076 (2019).

[4] Kopetzki, Anna-Kathrin, et al. "Evaluating Robustness of Predictive Uncertainty Estimation: Are Dirichlet-based Models Reliable?." International Conference on Machine Learning. PMLR, 2021.


**Summary Of The Paper:**

This paper mainly summarizes and evaluates the existing methods for estimating epistemic uncertainty through a single pass of the neural networks. The authors categorize deterministic uncertainty methods (DUMs) based on how latent representation is learned under regularization and how uncertainty is quantified. For evaluating DUMs, they provide some analyses on the uncertainty calibration performance under different-level distribution shifts for image classification and semantic segmentation. Specifically, they demonstrate that DUMs cannot generate well-calibrated uncertainty under distribution shifts compared to the MC-dropout and Ensemble methods.

**Summary Of The Review:**

Overall, this paper provides the evaluation for some determinist uncertainty estimation methods focusing on the robustness of uncertainty under distributional shifts. However, the paper should include more related methods in comparison with a discussion about the pros/cons and possible assumptions for each method. The experiment results should also be analyzed in detail.

---

> ### Author Response · Authors · 2021-11-15
> **Response to ZZs6**
>
> **Dirichlet-based methods**
>
> We thank the reviewer for highlighting these methods which we incorporated into our work. While a comparison against [1] would be unfair, since it requires training on OOD data, we agree that [2] and [3] can be beneficial for the analysis in this work. [2] demonstrates the best performance according to [4] and is more efficient than [3] which requires an ensemble for distillation at training time. Thus,  we incorporated [2] as a representative for Dirichlet-based methods on CIFAR10/100-C and added the results to the manuscript. Note that the performance on CIFAR100 in terms of test accuracy is worse since [2] requires low dimensional hidden representations (6 dimensions in the original work when training on CIFAR10). We were able to increase the dimensionality to 10. However, increasing the dimensionality even further renders the end-to-end trained normalizing flow unstable. Consequently, the model also did not converge on semantic segmentation.
>
> **Reasons for the observed issues and possible solutions.**
>
> We provide intuition regarding the weakness of methods that are based on estimating the distribution of hidden representations in the supplement (see section 6.1 & 6.2.2). We further provided an additional intuitive explanation in the conclusion.  However, the problematic calibration of DUMs found in this work requires a more thorough theoretical investigation which would be out of the scope of this work.
>
> **Discuss each deterministic uncertainty method in terms of the strengths/weaknesses of each method**
>
> While we do not discuss individual methods in detail, we describe the behaviour of entire categories in our taxonomy. Overall, we observe that methods based on distances or log-probabilities in the feature space are poorly calibrated. However, methods that rely on a discriminative approach (i.e. they derive their uncertainty from their predictive distribution), e.g. SNGP, can consistently be better calibrated than softmax. Further, in terms of types of regularization approaches we observe that approaches based on distance awareness fail to show a correlation between regularization strength and uncertainty quality, in particular also in terms of OOD detection which is what they were originally designed to do. We tried to emphasize these points more clearly in the conclusions of our manuscript and would invite further feedback from the reviewer.
>
> **Discussion about the OOD detection results**
>
> We do not emphasize our results on OOD detection (see supplement Tab. 4) since they do not contain new insights. These only confirm that DUMs are competitive in terms of OOD detection which has been already found by prior work. Note that it is interesting to observe that strong OOD detection does not necessarily imply well calibrated uncertainty. This hopefully serves as a helpful example for future evaluations of novel approaches to estimating epistemic uncertainty.
>
> **Experiments (1) Why do most DUMs have much worse calibration performance than softmax?**
>
> We stress that these DUMs are the ones that rely on distances or log-likelihoods of hidden representations for uncertainty estimation. In order to be well calibrated, these methods require that e.g. distances in the feature space entail information about expected model performance (which would be the case for a well calibrated uncertainty estimate). However, we empirically observe that this is not the case, or at least not easy to decoder, using any of these methods.
>
> **Experiments (2) Why do we only evaluate 3 DUMs on semantic segmentation?**
>
> We stress that these methods have previously not been evaluated on semantic segmentation. Thus, they do not naturally all scale well to such a task. In particular, DUQ, which also did not converge on CIFAR100 due to noisy centroids, did diverge on semantic segmentation. Moreover, DCU relies on expensive contrastive pretraining which makes it practically impossible to be applied to a large-scale dense prediction task.
>
> **Experiments (3) The experiments did not consider the regression tasks.**
>
> Unfortunately, almost none of the original works on DUMs considers regression tasks. Thus, we found it unsuitable for comparing this line of work on regression and outside of the scope of this work. We hope that future study will investigate deterministic uncertainty estimation methods for regression tasks.

---

> > ### Comment · Reviewer_ZZs6 · 2021-11-21
> > **Responces to the Authors**
> >
> > Thanks for the authors’ responses. We noticed that an additional explanation of the uncertainty estimation for DUMs is added to the supplementary materials, which may provide some thoughtful insights about the performance of each DUM. However, we still think it is necessary to talk about the weaknesses/strengths of each method in detail since it is mainly a review paper. Although the authors have taken the Posterior Network method into comparison in section 3.1.2, some of the illustrations are confusing. For example, why Posterior Network proposes to “maximize the entropy of normalizing flow during training”? Hence, we will not change the original score.

---

> > > ### Author Response · Authors · 2021-11-23
> > > **Response to ZZs6**
> > >
> > > We thank the reviewers for their feedback. We used this feedback to extend the supplement section 6.2 which now serves as a general overview of DUMs. This overview is split in two parts. The first part yields more details for the modularized perspective on DUMs taken in the main paper where we discuss the individual components found in all DUMs. We find this particular perspective very important since (1) many DUMs share components and are in general very similar and (2) it allows researchers and practitioners to distance themselves from particular methods and identify common components and trends, further providing assistance to the design choices for their method. We further added runtime/memory considerations for each regularization technique to this part. The second part denotes a method-centered perspective as requested by the reviewer. Here, we discuss each method in our empirical comparison. This section also includes a comparison of high-level characteristics that we deemed relevant for other researchers/practitioners.
> > >
> > > We emphasize however, that the main focus of this work remains the investigation of the calibration of DUMs under distributional shifts. Consequently, we argue that this additional material on DUMs is correctly placed in the supplement where a reader that is interested beyond the quality of predicted uncertainties produced by DUMs might consider it valuable information.
> > >
> > > **PostNet: “maximize the entropy of normalizing flow during training”**
> > >
> > > We apologize for the lack of explanation regarding this statement. We added a detailed explanation to the supplement and rephrased it as “entropy regularization” (see. supplement section 6.2.2). Essentially, we view posterior networks through the lens of this work which requires identifying their epistemic uncertainty estimation method (likelihoods of hidden representations) and their regularization technique to avoid feature collapse (entropy regularization of the distribution of hidden representations). The latter is achieved implicitly by regularizing the entropy of the predicted dirichlet distribution. This however implicitly encourages large entropies of the end-to-end learned normalizing flows. This can be understood by considering the entropy of a Dirichlet distribution and how the likelihood of the normalizing flows is used to parameterize the Dirichlet distribution as proposed in posterior networks (see supplement 6.2.2).
> > >
> > > Lastly, we do not discuss the Dirichlet distribution when discussing uncertainty estimation in Dirichlet-based approaches, since the particular parameterization of the output of the neural network (i.e. the Dirichlet distribution) becomes only relevant for aleatoric uncertainty. It does not impact the considerations in this work - i.e. epistemic uncertainty -  beyond the arguments on entropy regularization. Epistemic uncertainty in Posterior Networks is essentially derived from the likelihood provided by the normalizing flows (see definition of the parameters of the Dirichlet distribution in Posterior Networks). However, regularizing the entropy of the distributions of hidden representations implicitly - allowed by estimating the density end-to-end - denotes an interesting approach to preventing feature collapse and a valuable addition to our study.

---

### Official Review · Reviewer_z2yi · 2021-11-02

**Correctness:** 4
**Technical Novelty And Significance:** 4
**Empirical Novelty And Significance:** 3
**Recommendation:** 8
**Confidence:** 5

**Main Review:**

Strengths

- The motivation of the paper is strong, these new uncertainty quantification methods (DUMs) can raise some doubts as they do not do the usual bayesian model averaging (like ensembles or dropout/dropconnect), and as far as I know, there is no previous comparison of all these methods, particularly in more realistic settings. There is the important question if these methods work well in more complex settings.
- The authors propose a taxonomy on DUM method which I find useful for the future and very positive. The survey and description in the first pages of the paper is also very useful for the community.
- The evaluation seems to be correct, the dataset selection for image classification and segmentation experiments is excellent, and the metric selection is good, except for the rAULC metric which is not that common, but seems appropriate as not all evaluated methods produce probabilities. The authors argue why standard calibration errors cannot be used, but I think it could be an improvement to normalize confidence scores produced by DUMs to be able to use calibration errors, I do not see this as a big issue more than using a standard metric. For example gradient uncertainty can be normalized to produce scores in [0, 1] and ECE can be computed for these predictions.
- The supplementary material contains many additional results of interest, like variations of DUM hyper-parameter effects, justification about issues with intermediate activations, full results for corrupted versions of all datasets, and OOD detection without dataset shift (standard OOD detection benchmarks), showing that some DUMs work better in some datasets, and ensembles on SVHN.
- The additional segmentation results in the supplementary are very interesting, in particular Fig 7 and Fig 8, as it shows how MIR produces radically different uncertainty as seen in the heatmaps of uncertainty (last column), in comparison with other methods with mostly produce high uncertainty in the borders between regions, acting as a kind of edge detector, and this could lead to additional insights on why some methods fail.
- There are clear conclusions from this paper, that DUM methods do not perform OOD at acceptable levels for both classification and segmentation (except for DCU), and their calibration also suffers considerably. All of this in distribution shift settings.
- Figure 1 shows that DUMs suffer from the same issue with dataset distribution shift, where OOD/calibration performance suffers when corrupting input images, same as Ovadia et al. 2019.
- The paper is overall well written, and I had no trouble reading it.
- I find that the analysis in this paper is original and novel, there is small epsilon contribution to the state of the art, which should enable future research on DUMs.

Weaknesses

- A strong theoretical foundation on why these methods fail would strengthen the paper. The authors perform an analysis of regularization strength on MNIST (Easy but not many conclusions can be drawn from this dataset) and about distributions of intermediate features, but I think a full theoretical paper on this topic would be required.

Minor Issues

- It is possible that the paper would benefit from a "page 2 figure" to introduce the reader. There is no space, but I think putting Table 1 on top of the second page would serve this purpose.
- In Figure 1, I think axis tick labels can be reduced a bit in size or even factored across different plots (putting the tick labels in the first plots on the right and bottom rows) to save space and make the plots a bit bigger, for readability.
- One DUM that I think was left out is gradient-based uncertainty, for this please see " Classification uncertainty of deep neural networks based on gradient information" Oberdiek et al. 2018. I am not sure if adding it would change the conclusions, but maybe it can be included in the taxonomy.
- I totally agree with the authors that DUQ could diverge on CIFAR100-C, I have experienced this using DUQ on small datasets. I suggest that the authors remove the DUQ centroid learning algorithm (the running average over input features) and train the centroids using gradient descent. This will probably solve the issue.
- For readability, in Figure 6 of the supplementary, it is better to include the metric in the label of the y axis. Same for Figures 11 to 20, the reader will be pleased to find metrics and information in the x/y axis labels.

**Summary Of The Paper:**

This paper is an analysis and benchmark comparison of deterministic uncertainty quantification models under dataset shift. The authors evaluate DUM methods, and show that while they work well in academic datasets (mostly CIFAR10/100), they seem to fail in out of distribution detection on a more realistic setting (Semantic segmentation on CityScapes and CARLA simulated images) and in distribution shift settings (corrupting input images), particularly in terms of lower quality uncertainty calibration and out of distribution detection performance, which has practical implications for the use of these algorithms.


**Summary Of The Review:**

I believe that this is a good paper, with important conclusions and insights for the uncertainty quantification community, it is correctly evaluated, and should be accepted. There are no major issues to be dealt with.

---

> ### Author Response · Authors · 2021-11-15
> **Response to z2yi**
>
> **Theoretical foundation of the failure of some DUMs.**
>
> We agree with the reviewer that a deeper theoretical investigation of these works would be interesting and necessary to advance the field. While we aim to give some intuition in the supplement (section 6.1) regarding the limitations of DUMs that estimate uncertainty using the distribution of hidden representations, we understand that a more rigorous consideration of these approaches is needed. However, we note that an in-depth theoretical investigation is out of the scope of this work and requires its own work. Nevertheless, we expanded Sec. 6.2.2 with more theoretical background on the uncertainty estimation methods adopted by each DUM analysed and we provide intuitive explanations on why DUMs relying on uncertainty proxies based on the distribution of in-domain hidden features fail at being well-calibrated.
>
> **Applying ECE by squeezing uncertainty estimates into the interval [0, 1]**
>
> This is an interesting idea. However, this raises the question on how uncertainty values should be mapped into the interval [0, 1]. If we simply squeezed them linearly, we would assume that the quotient between uncertainty and probability is constant. Intuitively, it is very unlikely that this property emerges naturally for distances or log-likelihoods in the latent space and a non-linear mapping would be required.
>
> **Classification uncertainty of deep neural networks based on gradient information**
>
> We added this work to our related work section. However, it does not regularize its representations to be informative about its input. Thus, it likely would suffer from feature collapse. It moreover directly depends on aleatoric uncertainty, namely the softmax probability p. More precisely, -log(p) which is backpropagated. While it also depends on the values of intermediate representations due to the nature of backpropagation, it is unclear why the magnitude of these gradients should be zero in the case of low epistemic uncertainty. In particular, imagine an overconfident softmax prediction of which there are many (practical) examples. In this case the gradient magnitude would be small while epistemic uncertainty should be large.
>
> **Directly optimizing the class centroids in DUQ**
>
> We thank the reviewer for this suggestion. This is a promising approach to eliminate noisy centroids at training time. However, for a fair comparison of proposed DUMs we prefer not to alter the training procedure detailed in the original paper.

---

### Official Review · Reviewer_WCgZ · 2021-11-02

**Correctness:** 1
**Technical Novelty And Significance:** 2
**Empirical Novelty And Significance:** 2
**Recommendation:** 3
**Confidence:** 4

**Main Review:**

Title of the paper seems too broad and not representative of the content. “Practicality” is a very general term. The analysis and findings seem rather specific. It mainly seems to be that ”DUMs are not well calibrated under continuous distributional shifts”.

A comprehensive review of DUMs could be of great benefit, if done correctly. The area is a bit non-structured and the evaluations are not standardized. However, the current version of this paper falls short in helping in both aspects.

Literature review:
The explanations are very surface level and non-comprehensive. As an example, the literature review on the “discriminative methods” summarizes every approach in one very vague sentence, requiring the reader to refer to the papers to actually be able to understand. Given that the work is mostly a review of other methods, it should at least serve as a good survey, for a non expert reader. The current version of the manuscript falls short as is.

Besides that, the literature review of uncertainty estimation is incomplete (see [1, 2, 3, 4, 5] as examples). MIMO approaches [2, 3] are similar to deep ensembles but without the computational and memory requirements. One may argue that these approaches fall much closer to DUMs as opposed to Ensemble methods in terms of efficiency (memory & computation). Especially given that they are mostly image based, they are directly applicable here. They’re not even mentioned in the literature review.

The terms out-of-distribution detection, epistemic uncertainty estimation, aleatoric uncertainty estimation, and calibration. are not very well defined in the community when it comes to their quantification. Especially OOD detection (maybe the most well defined one) and epistemic uncertainty estimation, are often used interchangeably in many papers (debatable if correctly or incorrectly).
When it comes to adding synthetic corruption to the image (input), how is that not aleatoric uncertainty? One may argue that OOD detection is a better proxy for epistemic (although not perfect), and calibration on corruptions (noise in input) is better for aleatoric uncertainty. One example for each: [1] for using input variation / corruption for aleatoric uncertainty evaluation. [4] for using OOD detection as a proxy for epistemic uncertainty estimation. If that is the case, why should models designed for epistemic uncertainty estimation, be the best candidates for aleatoric uncertainty estimation anyway.

If the authors disagree with these conventions they should at least provide a comprehensive justification on why they think things should be different, and they should propose clear evaluation protocols for each type of uncertainty estimation.



[1] Modeling Uncertainty With Hedged Instance Embedding

[2] MixMo: Mixing Multiple Inputs for Multiple Outputs via Deep Subnetworks

[3] Training Independent Subnetworks For Robust Prediction

[4] Sketching Curvature for Efficient Out-of-Distribution Detection for Deep Neural Networks

[5] Epistemic Neural Networks


**Summary Of The Paper:**

This paper analyzes deterministic uncertainty estimation methods, in terms of their calibration under distributional shift. They provide a literature review, propose a quantification metric, and compare performance of several uncertainty estimation methods.

**Summary Of The Review:**

Overall I think the experiments of the paper are insightful and have merit, but the presentation, positioning, and claims could be significantly improved. Maybe if the paper’s focus, description, and claims were concentrated around “evaluating calibration of deterministic epistemic uncertainty models under distribution shift”, it would be more representative of the experiments, and it could become a more conclusive work.

---

> ### Author Response · Authors · 2021-11-15
> **Response to WCgZ**
>
> **Relation between practicality, calibration and OOD detection**
>
> The practicality of an epistemic uncertainty estimate is determined by two relevant properties - namely OOD detection and uncertainty calibration under distributional shift. We indeed focus on the calibration under continuous distributional shifts, since it obviously raises questions due to the lack of theoretical guarantees and empirical evidence of DUMs. Moreover, it is a relevant direction for practitioners and has previously not been investigated. However, we also provide results on OOD detection in the supplement. Note that these results do not yield new insights as they only confirm results of prior work - in particular, that DUMs denote a valid efficient alternative to more computationally expensive approaches (e.g. Deep Ensembles).
>
> **Unstructured field and non-standardized evaluation**
>
> We agree with the reviewer that the area of DUMs is currently lacking structure. To this end, we provide the first taxonomy of DUMs. In particular, we identify two axes that play a role in all DUMs (regularization technique and uncertainty estimation technique) and further subdivide both. Thus, we provide structure in an unstructured area. We apologize if the reviewer expected a more fine-grained taxonomy or introduction to existing works.While this would be beyond the scope of this work since our main focus lies on the empirical comparison of various DUMs, we extended the supplementary material with additional theoretical background on the possible design choices of DUMs and the different uncertainty estimation techniques adopted. Moreover, for each design choice we provided an explanation of the possible shortcomings. Please refer to Sec 6.2.2 of the Supplementary Material. For more details we refer to the original works. Regarding the evaluation protocol we argue that the metrics used in this work are well suited to become a standard in this line of work as they allow integrating methods that do not directly predict probabilities in the comparison.
>
> **Related literature**
>
> We stress that this work focuses on methods that aim at estimating epistemic uncertainty while treating the weights of a neural network deterministically. Since [2, 3] are efficient versions of ensembles and [4, 5] consider BNNs, they do not fall under this definition. Moreover, [1] considers aleatoric uncertainty. Broadening the scope would not allow us to provide an in-depth analysis of the shortcomings of current DUMs, which we believe necessary to foster development in the field. Nevertheless, we thank the reviewer for pointing us to these additional related works which we included in our related work section.
>
> **Terminology**
>
> We agree with the reviewer that the terms OOD detection and epistemic uncertainty are often used synonymously. In particular, we also observe this in the area of DUMs where methods are often exclusively evaluated on OOD detection. This denotes the main motivation for our empirical investigation since good OOD detection does not necessarily imply good epistemic uncertainty. Good epistemic uncertainty also requires the uncertainty estimate to hold information about how likely the model made a mistake, i.e. how well it is calibrated.
>
> **Synthetic corruptions**
>
> We note that we are not the first to use synthetic corruptions for evaluating epistemic uncertainty [24]. Aleatoric uncertainty denotes the uncertainty present in the data itself. Thus, for corruptions that do not leave the training data distribution, the reasoning of the reviewer holds true. However, the type and magnitude of the synthetic corruptions adopted in our study is chosen such that they are not observed in the training data. Consequently, this type of noise cannot be learned at training time (e.g. by learning to predict categorical distributions of higher entropy). Thus, it is necessary to take into account epistemic uncertainty to correctly model such corruptions. We tried to clarify this point more extensively in the introduction.

---

> > ### Comment · Reviewer_WCgZ · 2021-11-30
> > **Reponse to Authors Rebuttal**
> >
> > The author's response and updated manuscript further reinforces my impression and does not refute any of the main points. Thus my main 2 concerns are remaining unaddressed.
> >
> > Concern 1: The work is very specific (calibration under distribution shift), but claims are very general (“practicality” of deterministic epistemic uncertainty).
> >
> > Authors: “The practicality of an epistemic uncertainty estimate is determined by two relevant properties - namely OOD detection and uncertainty calibration under distributional shift. “
> >
> > Response: This is not an established definition. The authors are defining it as such.
> >
> > Authors: “We indeed focus on the calibration under continuous distributional shifts.”
> >
> > Response: Then this work should be called calibration under distribution shift.
> >
> > The authors are defining a subjective definition “practicality of epistemic uncertainty”, which sounds larger in scope and more general. And then mainly focus on one very specific aspect of it. This seems like an unnecessary generalization of the scope, which could mislead a novice reader's conclusions, especially given the incompleteness of the work (please see concern 2).
> >
> >
> > Concern 2: Image Corruption/manipulation: epistemic vs aleatoric.
> >
> > The authors claim that: “the type and magnitude of the synthetic corruptions adopted in our study is chosen such that they are not observed in the training data”. How is this possibly ensured/verified? The fact that this statement cannot be empirically verified, is the reason that there are dissenting opinions on corruption/manipulations.
> >
> > Authors: “We note that we are not the first to use synthetic corruptions for evaluating epistemic uncertainty [24]. “
> >
> > Response: You are not refuting my point. Some people call that epistemic (such as [24]), some people call that aleatoric(such as [1]). There are different opinions. So they should all be at the very least discussed in a survey work.
> >
> > Authors: “Moreover, [1] considers aleatoric uncertainty. ”
> >
> > Response: Seems like the authors agree that corrupting/manipulating an image could be aleatoric uncertainty then.
> >
> > Authors: “Thus, we provide structure in an unstructured area. ”
> >
> > Response: It is an incomplete structure, which reinforces one side of the arguments (referring to the previous 2 comments). In a survey work, this could paint an incomplete picture for the reader. Setting structure in an unstructured area requires at least mentioning the different opinions on the topic and discussing pros and cons of each.
> >
> >
> > I still think this, if presented as an analysis of DUMs under distribution shift, would have merit, but not with the current presentation and claims.

---

> > > ### Author Response · Authors · 2021-11-30
> > > **Reponse to WCgZ**
> > >
> > > **Response: This is not an established definition. The authors are defining it as such.**
> > >
> > > To the best of our knowledge, there is no prior definition of the practicality of an epistemic uncertainty estimate. Furthermore, we argue that calibration and OOD detection performance, which we both investigate in this work, are the only components that determine the usefulness, i.e. practicality, of an epistemic uncertainty estimate. Thus, we do not see any problems with our title and scope.
> > >
> > > **Response: Then this work should be called calibration under distribution shift.**
> > >
> > > We focus on the calibration under distributional shifts since it denotes a vastly underexplored element of DUMs. However, we also provide experiments on OOD detection (see supplement). Thus, the term “practicality” in our title is justified, since we holistically investigate DUMs regarding their usefulness in practice.
> > >
> > > **Concern 2: Image Corruption/manipulation: epistemic vs aleatoric.
> > > The authors claim that: “the type and magnitude of the synthetic corruptions adopted in our study is chosen such that they are not observed in the training data”. How is this possibly ensured/verified? The fact that this statement cannot be empirically verified, is the reason that there are dissenting opinions on corruption/manipulations.**
> > >
> > > This statement is true by construction and has also been verified empirically. In particular, we refer the reviewer to the original work proposing the use of artificial corruptions [1]. This set of corruptions has particularly been designed to analyze generalization properties - i.e. performance when leaving the data distribution - of neural networks. Furthermore, the corruptions in our custom dataset collected in CARLA also do not exist in the training dataset, which is straightforward to achieve in a simulation that allows fine-grained control over the environment parameters.
> > >
> > > **Response: You are not refuting my point. Some people call that epistemic (such as [24]), some people call that aleatoric(such as [1]). There are different opinions. So they should all be at the very least discussed in a survey work.**
> > >
> > > The corruptions used in our evaluation are leaving the training data distribution by construction. Furthermore, following the predominant definition of aleatoric uncertainty [2, 3, 4],  aleatoric uncertainty encopasses the uncertainty present in the training data. Consequently, since we leave the training data by construction with aforementioned corruptions, we argue that the perspective taken in this paper resonates with the current state of the research community.
> > >
> > > **Authors: “Moreover, [1] considers aleatoric uncertainty. ”
> > > Response: Seems like the authors agree that corrupting/manipulating an image could be aleatoric uncertainty then.**
> > >
> > > This conclusion is taken out of context. We are saying that [1] (WCgZ’s citations) quantifies aleatoric uncertainty. We do not say that corruptions that leave the training data distribution are optimally detected by aleatoric uncertainty.
> > >
> > > **Authors: “Thus, we provide structure in an unstructured area. ”
> > > Response: It is an incomplete structure, which reinforces one side of the arguments (referring to the previous 2 comments). In a survey work, this could paint an incomplete picture for the reader. Setting structure in an unstructured area requires at least mentioning the different opinions on the topic and discussing pros and cons of each.**
> > >
> > > Firstly, providing a survey is not the main motivation of this work, but rather investigating the usefulness of a recently emerging set of methods for uncertainty estimation. Nevertheless, we provided extensive additional details on the underlying methods in the supplement. Secondly, the incomplete structure mentioned by WCgZ refers to a topic outside of DUMs. The main focus of this work are DUMs and not the interpretation of corrupted data in the light of aleatoric and epistemic uncertainty (as suggested by the reviewer by referring to the previous two comments).
> > >
> > > **I still think this, if presented as an analysis of DUMs under distribution shift, would have merit, but not with the current presentation and claims.**
> > >
> > > We are glad that the reviewer finds the conclusions in this paper beneficial for the research community.
> > >
> > > [1] Hendrycks, D. and Dietterich, T., 2018, September. Benchmarking Neural Network Robustness to Common Corruptions and Perturbations. In International Conference on Learning Representations.
> > >
> > > [2] Kiureghian, A. D. and Ditlevsen, O. Aleatory or epistemic? does it matter? Structural Safety, 31(2):105–112, March 2009. ISSN 0167-4730. doi: 10.1016/j.strusafe.2008.06. 020.
> > >
> > > [3] Gal, Y. 2016. Uncertainty in deep learning. University of Cambridge, 1: 3.
> > >
> > > [4] Kendall, A.; and Gal, Y. 2017. What uncertainties do we need in bayesian deep learning for computer vision? In Advances in neural information processing systems, 5574–5584.

---

### Author Response · Authors · 2021-11-15
**General Comment**

We thank the reviewers for their detailed and constructive feedback and updated the manuscript accordingly.

In particular, we provided additional results using a representative for Dirichlet-based approaches (Posterior Networks) on CIFAR10/100-C as suggested by ZZs6. Moreover, we updated Section 6.2 of the supplementary material and extended it with additional theoretical background on DUMs. Further, we provided more qualitative explanations of shortcomings of DUMs in our conclusion and Section 6.2.2.

---

> ### Author Response · Authors · 2021-11-23
> **General Comment regarding the Response of ZZs6**
>
> Following the suggestions of ZZs6, we added additional discussion regarding the strengths and weaknesses of each DUM in the supplementary material. We would like to direct the attention of the reviewers to our response to ZZs6.

---

### Decision · Program_Chairs · 2022-01-20

**Decision:**

Reject

**Comment:**

The paper performs an empirical evaluation of deterministic methods for the quantification of epistemic uncertainty.  There is no new algorithm.  The main contribution is the empirical evaluation.  This empirical evaluation will be useful for the community. It is an independent evaluation that casts some doubts on the calibration of several existing deterministic techniques, which will spur additional research. However, the paper is not well written. As pointed out by the reviewers, the paper does not provide much background. It refers to many  concepts without defining them. The concepts are not new (references are provided for each concept), but since the paper does not describe any new technique it should do a good job at explaining those concepts. The authors added some explanations in the supplementary material, but some of those explanations should really be in the main paper.  The most important issue with the paper is that it does not explain why the deterministic techniques do not seem to be well calibrated. The authors added a "theoretical justification" in section 6.1, but it amounts to saying that deterministic methods make a point estimate, which is too general to explain much. An important factor for proper generalization and calibration is the inductive bias of the model. At the end of the day, if we generate data from a model, then that model will be better calibrated than the other models. So a discussion of the inductive bias of each model and how this inductive bias relates to the properties of each dataset would have been much more insightful.